# Statins Inhibit the Gliosis of MIO-M1, a Müller Glial Cell Line Induced by TRPV4 Activation

**DOI:** 10.3390/ijms23095190

**Published:** 2022-05-06

**Authors:** Youn Hye Jo, Go Woon Choi, Mi-Lyang Kim, Kyung Rim Sung

**Affiliations:** 1Department of Ophthalmology, Konkuk University Medical Center, College of Medicine, Konkuk University, 120, Neungdong-ro, Gwangjin-gu, Seoul 05030, Korea; 15dbsp@naver.com; 2Department of Ophthalmology, Asan Medical Center, College of Medicine, University of Ulsan, 88, Olympic-Ro 43-Gil, Songpa-gu, Seoul 05505, Korea; 3Biomedical Research Center, Asan Medical Center, College of Medicine, University of Ulsan, 88, Olympic-Ro 43-Gil, Songpa-gu, Seoul 05505, Korea; goni8586@gmail.com (G.W.C.); iota76@gmail.com (M.-L.K.)

**Keywords:** glaucoma, Müller gliosis, NF-κB pathway, statin, TNF-α

## Abstract

We characterized Müller cell gliosis induced by the activation of transient receptor potential vanilloid-type 4 (TRPV4) and assessed whether statins could modulate the gliosis. The human Müller cell line, MIO-M1, was used to analyze the gliosis caused by glaucomatous stimulation. To induce Müller gliosis in MIO-M1 cells, GSK101 was used to activate TRPV4, and Müller gliosis was evaluated by analyzing vimentin, nestin, and glial fibrillary acidic protein (GFAP) expression. The expression level of TNF-α was determined by ELISA. To evaluate the GSK101 activation of the NF-κB pathway, p65 phosphorylation was measured by Western blotting, and the nuclear translocation of p65 and IκBα phosphorylation were assessed by immunostaining. To assess the effect of statins on MIO-M1 gliosis, cells were pretreated for 24 h with statins before GSK101 treatment. Vimentin, nestin, and GFAP expression were upregulated by GSK101, while statins effectively inhibited them. The expression of TNF-α was increased by GSK101. The phosphorylation and nuclear translocation of p65 and IκBα phosphorylation, which occurs prior to p65 activation, were induced. Statins suppressed the GSK101-mediated phosphorylation of IκBα and p65 translocation. Statins can mitigate gliosis in the human Müller cell line. Because TRPV4 activation in Müller cells reflects glaucoma pathophysiology, statins may have the potential to prevent RGC death.

## 1. Introduction

Glaucoma is a progressive neurodegenerative eye disease characterized by irreversible structural damage to the retinal ganglion cells (RGCs) [1,2]. RGCs play critical roles in the transfer of visual information from the retina to the brain via their axons, i.e., the retinal nerve fiber layer (RNFL). In glaucoma, axonal degeneration and the apoptotic cell death of RGCs ultimately lead to irreversible vision loss. While lowering intraocular pressure (IOP) indirectly protects RGCs, there is no proven glaucoma treatment that directly targets RGCs. Therefore, understanding the mechanism of RGC dysfunction is important for the development of new glaucoma treatment strategies.

RGCs receive physiological and pathological support from glial cells, such as astrocytes and Müller cells. Healthy Müller cells are specialized radial glial cells that span the entire retina and provide metabolic support. However, changes in Müller cells are involved in almost all types of retinal degeneration, such as age-related macular degeneration, diabetic retinopathy, and glaucoma, as well as normal retinal function [3,4]. When the surrounding tissues are damaged by increased IOP or inflammation, Müller cells induce neuronal degeneration and edema by gliotic alteration and may form glial scars, which interfere with the repair and regeneration of retinal tissues [3,5]. In addition, Müller gliosis is induced by glaucomatous stimulation and implicated in RGC apoptosis. Tumor necrosis factor-alpha (TNF-α), a detrimental cytokine, is mainly synthesized and released by activated Müller glial cells and induces RGC apoptosis [6,7,8,9,10,11]. Thus, it can be postulated that the suppression of Müller gliosis is neuroprotective. Studies of Müller cell function in vitro are challenging because the cells tend to rapidly differentiate in culture, making it difficult to isolate pure cell populations. In 2002, MIO-M1 was reported as the first spontaneously immortalized human Müller cell line [12]. Thereafter, MIO-M1 cell lines were used in several in vivo studies [13,14] for evaluating Müller cells. We also used MIO-M1 cells in this study.

Because the transient receptor potential vanilloid-type 4 (TRPV4) channel contributes to the resting membrane potential of Müller cells, its specific agonist, GSK1016790A (GSK101), was used to activate gliosis [15,16]. Then, vimentin [17], nestin [18], and glial fibrillary acidic protein (GFAP) [19,20] were used to verify the induction of Müller gliosis. Because GSK101 is a candidate for inducing gliosis through TNF-α, we assayed TNF-α levels in response to GSK101 and monitored how they changed after statin treatment.

NF-κB (p50/p65), an inducible nuclear transcription factor, regulates inflammatory factors, including TNF-α [21,22]. NF-κB is inactive when bound to IκB (inhibitor κB). When NF-κB is released from IκB, it is free to translocate to the nucleus, where it acts as a transcription factor. Thus, p65 and phospho-p65 were assayed in the nucleus and cytosol, respectively, after GSK101 and statin treatments. IκB kinase (IKK) complex phosphorylation occurs in response to upstream signals after the activation of the IκB kinase complex (IκBα/IκBβ/IκBγ). Active IKK induces IκB degradation, and NF-κB is released. Thus, we analyzed the concentration of phospho-IκBα by immunostaining after GSK101 and statin pretreatments. 

Statins, widely used as hyperlipidemia treatments, not only regulate cholesterol biosynthesis but also exert pleiotropic effects, such as fibrosis, inflammation, and immune response control [23,24,25]. Statins’ neuroprotective effects have been reported in many central nervous system (CNS) diseases, such as ischemic stroke, Alzheimer‘s disease, and multiple sclerosis [26]. Therefore, we expected statins would have beneficial effects on the optic nerve and RNFL. In fact, several previous studies have indicated an association between glaucoma and statins [27,28,29,30,31,32,33,34]. As clinical evidence about the protective effect of statins on glaucoma development and progression has increased, in vitro studies have also been published. The secreted protein, acidic and rich in cysteine (SPARC), and the extracellular matrix (ECM) protein have emerged as major modulators of IOP through changing the outflow facility, and lovastatin inhibits the expression of TGF-β2-mediated SPARC in human trabecular meshwork cells [35]. In addition, we reported that statins exert antifibrotic effects by inhibiting the expression of TFG-β2-mediated ECM molecules in astrocytes of the human optic nerve head (ONH) [36,37]. However, no research about the direct effects of statins on Müller glial cells as protection for dysfunctional RGCs has been conducted.

In this paper, we assessed whether Müller gliosis could be induced by the activation of TRPV4, a mechanosensor that may respond to increased IOP. To do this, we evaluated the expression of gliosis markers, TNF-α expression, and the activation of the NF-κB pathway. Then, we determined whether statins could modulate TRPV4-mediated Müller gliosis.

## 2. Results

### 2.1. Evaluation of Müller Gliosis in MIO-M1 Cells and the Effect of Statins 

To evaluate Müller gliosis by TRPV4 activation, the expression of the Müller gliosis indicators vimentin, nestin, and GFAP was analyzed by immunostaining. Treatment with the TRPV4 agonist GSK101 upregulated vimentin, nestin, and GFAP expression (Figure 1A–C). Pretreatment with simvastatin, atorvastatin, and lovastatin effectively suppressed the induction of gliosis marker proteins. 

### 2.2. TNF-α as a Candidate to Induce Gliosis and Its Reduction by Statins

Then, we asked how the response of Müller gliosis to TRPV4 activation leads to detrimental effects on RGC. The most plausible candidate is TNF-α, because it can induce RGC apoptosis [6,7,8,9,10,11,38]. Therefore, we asked whether TNF-α production increases in GSK101-treated MIO-M1 cells. The cells were treated with GSK101 concentrations ranging from 1 to 50 nM. TNF-α expression was significantly induced at GSK101 concentrations ≥ 10 nM (Figure 2A). We also tested whether statins could suppress GSK101-induced TNF-α expression. With 50 nM GSK101, TNF-α production was induced 14-fold more than with 1 nM GSK101. Pretreatment with statins suppressed TNF-α production, even at concentrations as low as 1 nM (Figure 2B–D).

### 2.3. NF-κB Pathway in MIO-M1 Cells 

Because we confirmed that TRPV4 activation induces NF-κB activation and TNF-α secretion, and statins inhibit these events, we determined which step of the NF-κB pathway was inhibited to suppress TNF-α expression and consequently protect RGCs. NF-κB forms homo- or hetero-dimers with DNA-binding proteins. p65 is a major component of these dimers; when NF-κB is activated, p65 is phosphorylated, and the complex translocates to the nucleus. Therefore, we assayed p65 phosphorylation in cytosolic and nuclear fractions with GAPDH and lamin B1 as respective internal controls. We observed the robust induction of p65 phosphorylation in both the cytosolic and nuclear fractions by treatment with GSK101 in a dose-dependent manner (Figure 3). These data suggest that TNF-α induction by GSK101 is mediated by the NF-κB pathway in MIO-M1 cells.

### 2.4. Statins Inhibit the NF-κB Pathway

Because statins can suppress TNF-α induction, we tested whether they could suppress GSK101-mediated p65 phosphorylation. Treatment with 25 nM GSK101 consistently induced p65 phosphorylation in cytosolic and nuclear fractions (Figure 4A–C). Pretreatment with 0.2 nM of any of the statins significantly decreased p65 phosphorylation in the nucleus, and pretreatment with 0.2 nM simvastatin, 1 nM atorvastatin, and 10 nM lovastatin decreased p65 phosphorylation in the cytosol (Figure 4A–C). 

### 2.5. Statins Reduce IκBα Phosphorylation

To confirm that statins suppress the NF-κB pathway, we assessed the phosphorylation of the inhibitory protein IκBα by immunocytochemistry. IκBα functions as an inhibitor through interaction with the NF-κB complex. However, IκBα phosphorylation by stimulatory signals promotes IκBα ubiquitination and degradation, which induces NF-κB complex activation. Consistent with Figure 3 and Figure 4, GSK101 treatment increased the immunoreactivity for phospho-IκBα in the cytoplasm and nucleus. Treatment with statins prior to GSK101 treatment suppressed IκBα phosphorylation. All three statins were potent attenuators of GSK-mediated IκBα phosphorylation upstream of p65 activation in MIO-M1 cells (Figure 5).

## 3. Discussion 

Glaucoma can cause blindness due to progressive, irreversible optic nerve damage. The major risk factor for glaucoma is increased IOP, which is the only adjustable factor to modify glaucoma development and progression. Proposed mechanisms linking RGC injury and IOP elevation include the compressive effect of IOP on the lamina cribrosa [39], pressure-induced tissue ischemia [40,41], and local cellular inflammatory mechanisms [42]. Although studies have elucidated how glaucoma develops and progresses, there are no proven, commercial RGC-protecting and -regenerating glaucoma treatments, besides IOP reduction, to minimize stress on the ONH. This study is noteworthy because it suggests the use of statins as novel glaucoma treatment agents with neuroprotective effects.

Statins are prescribed for hyperlipidemia treatment; they lower blood cholesterol by competitively inhibiting HMG-CoA reductase, the rate-limiting enzyme of the mevalonate pathway. Since statins are structurally similar to HMG-CoA, the active site of the enzyme competes with the native substrate. This competition reduces the rate at which mevalonate is produced, the molecule needed to synthesizes cholesterol. In addition, the inhibition of HMG-CoA by statins hampers the downstream synthesis of isoprenoids, which are crucial lipid attachments for intracellular signaling molecules, such as Rho, Rac1, and CDC42 [43,44]. The suppression of protein prenylation is thought to be involved in the modulation of immune function, the improvement of endothelial function, and other pleiotropic cardiovascular benefits [43,45,46,47]. 

Considerable clinical evidence indicates that statins have beneficial effects on glaucoma patients. De Castro et al. [29] showed that statins may slow the progression of suspected glaucoma as measured by confocal scanning laser ophthalmoscopy. McGwin et al. [28] demonstrated that the prevalence of open-angle glaucoma (OAG) was lower in patients who had taken statins for more than 24 months. Additionally, Leung et al. [48] have reported that simvastatin may stabilize visual field progression. In a retrospective longitudinal cohort analysis, statin use was associated with a significant reduction in OAG risk [27]. A systemic review and meta-analysis indicated that short-term statin use was associated with reduced glaucoma incidence, but the effect of statins on glaucoma progression and IOP was unclear [49]. Furthermore, a large cohort study suggested that statin use significantly reduced OAG risk in hyperlipidemic patients [27]. Our recent study also showed that statins aided RGC survival by inhibiting ECM remodeling of the ONH [37]. As clinical evidence for the protective effect of statins on OAG development and progression has accumulated, in vitro studies on astrocytes and trabecular meshwork cells have also been published [35,37,50]. However, this study is the first to elucidate the protective effect of statins against glaucomatous stimuli in Müller glial cells. 

Müller glia play crucial roles in RGC survival, and Müller gliosis contributes to glaucoma pathogenesis [51,52]. Müller glia span the entire retina and have neuroprotective, as well as detrimental, effects in response to glaucoma-related neuronal injury via signaling cascades [4,5,53,54,55,56,57]. After retinal insults, Müller glia undergo reactivation, so-called Müller gliosis, characterized by the increased expression of intermediate filaments, such as vimentin, nestin, and GFAP [53]. TRPV4 is a membrane calcium channel activated by osmotic or mechanical pressure-related insults, causing calcium flow into the cell, increasing the intracellular calcium ion concentration [58,59]. TRPV4 induces gliosis and neurodegeneration in the brain [60,61] and is expressed in the retina, specifically in Müller glia. TRPV4 activation appears to be involved in RGC apoptosis. In a study using dissociated RGC cultures, RGC apoptosis and Müller gliosis increased due to TRPV4-agonist induced TRPV4 activation [62,63]; in a study using porcine retinal explants treated with the TRPV4-specific antagonist RN-1743, RGC survival increased [64]. Therefore, the mechanism of inflammation induction by TRPV4 has received attention. We used MIO-M1 cells with GSK101 as a TRPV4 channel agonist: the expression of each intermediate filament, vimentin, nestin, and GFAP, increased in MIO-M1 cells treated with 25 nM GSK101 and decreased by treatment with 10 nM statins (Figure 1). That is, we found that TRPV4 and statins induce and reduce the Müller gliosis of MIO-M1 cells caused by any mechanism, respectively.

We hypothesized that TNF-α production in Müller glia is involved in this mechanism because TRPV4 activation increases the secretion of proinflammatory cytokines, including TNF-α, from glia in other cells [65]. Although some studies have shown that increased IOP activates TRPV4 [66,67], it is not well-established that TRPV4 activation in Müller cells is linked with TNF-α production and glaucoma pathology. We found that TNF-α was induced by GSK101 in MIO-M1 cells (Figure 2). Therefore, this study provides evidence that TRPV4 is closely related to glaucoma pathophysiology through Müller gliosis and TNF-α production.

TNF-α is a well-known proinflammatory cytokine with multiple functions in the immune response, and its expression is induced by transcription factors in the NF-κB pathway (p50–p65 complex) [68,69,70]. It is mainly synthesized and released by activated glial cells and is involved in glaucomatous neurodegeneration; thus, it is considered a therapeutic target for improving neuroprotection [2,6,8,9,10,11,71,72,73]. In animal models, increased IOP, increased TNF-α in Müller glia [74,75], and Müller gliosis resulted in increased TNF-α expression [76]. TNF-α was also increased in the retinas of glaucoma patients [77,78,79]. Lebrun-Julien et al. [75,80] have reported that the intravitreal injection of TNF-α reduces the RGC numbers and causes optic nerve degeneration. Furthermore, TNF-α inhibitors prevent Müller gliosis and retinal ganglion cell loss [81,82]. We found that TNF-α production was increased by TRPV4 activation in MIO-M1 cells and decreased by statins, implying that statins could be therapeutic agents with neuroprotective effects for glaucoma patients, if proven by animal models, which will be our next research project.

We also confirmed that TNF-α induction by GSK101 in MIO-M1 cells is mediated by the NF-κB pathway, which statins inhibit by attenuating IκBα phosphorylation. These findings agree with previous studies [73,83,84]. Shi et al. [73] demonstrated that excitotoxic damage leads to selective NF-κB activation, which induces TNF-α production in Müller glial cells. Ahn et al. [83] showed that statins have a role in overcoming chemoresistance through the modulation of NF-κB in human myeloid leukemia KBM-5 cells. In addition, Tu et al. [84] demonstrated that simvastatins inhibit the NF-κB pathway by suppressing IκBα phosphorylation and degrading the nuclei of pulposus cells. However, this is the first study to reveal the activation of TNF-α through activated TRPV4 in Müller glia, its inhibition by statins, and the effects statins exert on the NF-κB pathway.

This study has some limitations. First, we used a Müller cell line, MIO-M1, rather than primary Müller cells. Therefore, in our next study, we will check whether the same action occurs in primary Müller cells through primary Müller cell culture or in vivo and, furthermore, in animal experiments. Numerous studies in the field of neuroinflammation have shown the potential of statins [85,86], but there are also some reports concluded that statins are ineffective [87,88]. Second, although TNF-α induces RGC apoptosis as a proinflammatory cytokine, the present study did not show that RGC apoptosis occurs when TRPV4 is activated. In future experiments, we will need to examine RGC apoptosis in vivo or in a Müller–RGC co-culture model. Third, we only examined Müller gliosis caused by TRPV4 activation, but TRPV4 is also expressed in RGCs and bipolar cells and can change with increased IOP [58]. Therefore, the effect of statins on TRPV4 activation in RGCs and bipolar cells should be investigated. Finally, NF-κB pathway activation is one way that TRPV4 signaling occurs. TRPV4 signaling increases intracellular Ca^2+^ concentration, which increases calpain [89]. There is a report that statins inhibit calpain activation [89]; therefore, the effects of TRPV4 activation on Muller or RGCs cannot be limited to the NF-κB pathway. It will be interesting to examine calpain activity on statins.

In conclusion, TRPV4 activation reiterates glaucomatous stimulation by GSK101 in human MIO-M1 cells, increasing TNF-α expression, and statins inhibit this expression by interfering with IκBα phosphorylation. Numerous studies have shown that TNF-α signaling is involved in glaucomatous neurodegeneration and is thus a possible treatment target to improve neuroprotection [2,71,72,73]. This understanding of the TNF-α expression pathway caused by glaucoma stimulation in Müller cells, and the reduction in TNF-α expression by statins, suggest statins as potential therapeutic agents for glaucoma treatment. This study has revealed an important aspect of statins’ protective effects in reactive Müller gliosis.

## 4. Materials and Methods 

### 4.1. Cell Culture

MIO-M1, a human Müller glial cell line, was grown in GlutaMAX^TM^ DMEM medium supplemented with 50 U/mL penicillin, 50 g/mL streptomycin (Invitrogen-Gibco-Life Science Technology, Karlsruhe, Germany) and 10% fetal bovine serum (FBS; Invitrogen-Gibco-Life Science Technology) [90]. MIO-M1 cells were maintained at 37 °C, 5% CO_2_ in a humidified incubator until the cells reached 80% confluency. The cells were detached by treatment with TrypLE (Invitrogen-Gibco-Life Science Technology, Waltham, MA, USA), and the resulting cell suspension was protected with a neutral trypsin solution (Invitrogen-Gibco-Life Science Technology) [91].

### 4.2. Reagents

For investigation of the NF-κB pathway, IκBα, p65, and phospho-p65 antibodies were obtained from Cell Signaling Technology (Danvers, MA, USA). Vimentin, nestin, and GAPDH antibodies were obtained from Santa Cruz Biotechnology, Inc. (Santa Cruz, CA, USA). The Human Quantikine TNF-α ELISA kit was obtained from R&D Systems (Minneapolis, MN, USA). The GFAP antibody and RIPA Cell Lysis buffer were purchased from ThermoFisher Scientific (Fair Lawn, NJ, USA). GSK1016790A (GSK101) was purchased from Sigma-Aldrich (St. Louis, MO, USA).

### 4.3. Immunofluorescent Staining of Human Müller Glial (MIO-M1) Cells

MIO-M1 cells were seeded on 12 mm glass coverslips placed in 24-well culture plates in growth medium. The cells were fixed with 4% paraformaldehyde and permeabilized with 0.1% Triton X-100. After three washes with phosphate-buffered saline (PBS), the cells were incubated for 1 h at 37 °C with primary antibodies (1:50) in PBS with 1% bovine serum albumin (BSA). After washing three times in PBS for 5 min each, the cells were incubated for 1 h at room temperature with a secondary antibody (Cy-3, Alexa594) (1:100) in PBS. The gliosis of MIO-M1 cells was confirmed through positive staining for vimentin, nestin, and GFAP. Cells were incubated with 1 μg/mL 4′,6-diamidino-2-phenylindole dihydrochloride (DAPI; Invitrogen-Gibco-Life Science Technology) to stain the nuclei. The coverslips were mounted with FluorSave (Calbiochem, San Diego, CA, USA), and fluorescent images were captured using an inverted microscope (Carl Zeiss Microscopy GmbH, Jena, Germany) and analyzed with Zeiss ZEN imaging software (Carl Zeiss Microscopy GmbH).

### 4.4. Separation of Nuclei and Cytosol 

MIO-M1 cells were seeded in 6-well culture plates. Cells were pretreated with statins followed by 25 nM of GSK101, a TRPV4 agonist. Nuclei and cytosols were separated using the Pierce^TM^ NE-PER Nuclear and Cytoplasmic Extraction Reagents (ThermoFisher Scientific) according to the manufacturer’s instructions. Briefly, cell lysis was demonstrated using the CER-I and CER-II lysis buffers of the NE-PER kit. The tube containing CER-I was centrifuged for 5 min at ~16,000× *g* in a microcentrifuge and the supernatant (cytosol) was transferred to a new Eppendorf tube. After suspending the insoluble fraction, ice-cold NER was added, and the cells were incubated for 40 min. After adding CER-II lysis buffer, centrifugation took place for 10 min at 16,000× *g*, the supernatant (nuclei) was transferred to a new Eppendorf tube, and each sample was analyzed by Western blotting.

### 4.5. Western Blot Analysis

Each cytosolic and nuclear sample was separated by SDS-polyacrylamide gel electrophoresis (PAGE) and transferred to nitrocellulose membranes (Cytiva, Marlborough, MA, USA), which were blocked in BSA for 1 h. The blots were incubated with antibodies against phospho-IκBα (p-IκBα), GFAP, and GAPDH (Santa Cruz Biotechnology, Inc.); p65 (Cell Signaling Technology); and phospho-p65 (p-p65; Invitrogen) in blocking solution overnight at 4 °C. The membranes were washed with PBST and then incubated with HRP-conjugated secondary antibodies for 2 h at room temperature. The membranes were developed using the Enhanced Chemiluminescence (ECL) detection system (Santa Cruz Biotechnology, Inc.). Band densities were quantified using FUJIFILM Science Lab Image Gauge Ver. 4.0 (Fuji Photo Film Co., Ltd., Tokyo, Japan).

### 4.6. Quantification of TNF-α

MIO-M1 cells were cultured in 6-well plates and then treated with GSK101 for 24 h or statins for 1 h followed by GSK101. The MIO-M1 cell culture medium was analyzed for TNF-α using an ELISA kit (R&D Systems, Minneapolis, MN, USA) according to the manufacturer’s instructions [92]. The absorbance at 450 nm was determined by an automated microplate reader (Vmax; Molecular Devices, Palo Alto, CA, USA), and the results were analyzed using Prism 6.05 (GraphPad Software, Inc., San Diego, CA, USA). 

### 4.7. Statistical Analysis

All values are expressed as means ± standard deviations (SD). The results represent three separate experiments conducted under the same conditions. Unpaired *t*-tests and one-way ANOVAs were used for statistical comparison. All statistical analyses were performed using an ANOVA test with Prism 6.05 (GraphPad Software, Inc.). * *p* < 0.05 and ** *p* < 0.001 were considered significant.

## Figures and Tables

**Figure 1 ijms-23-05190-f001:**
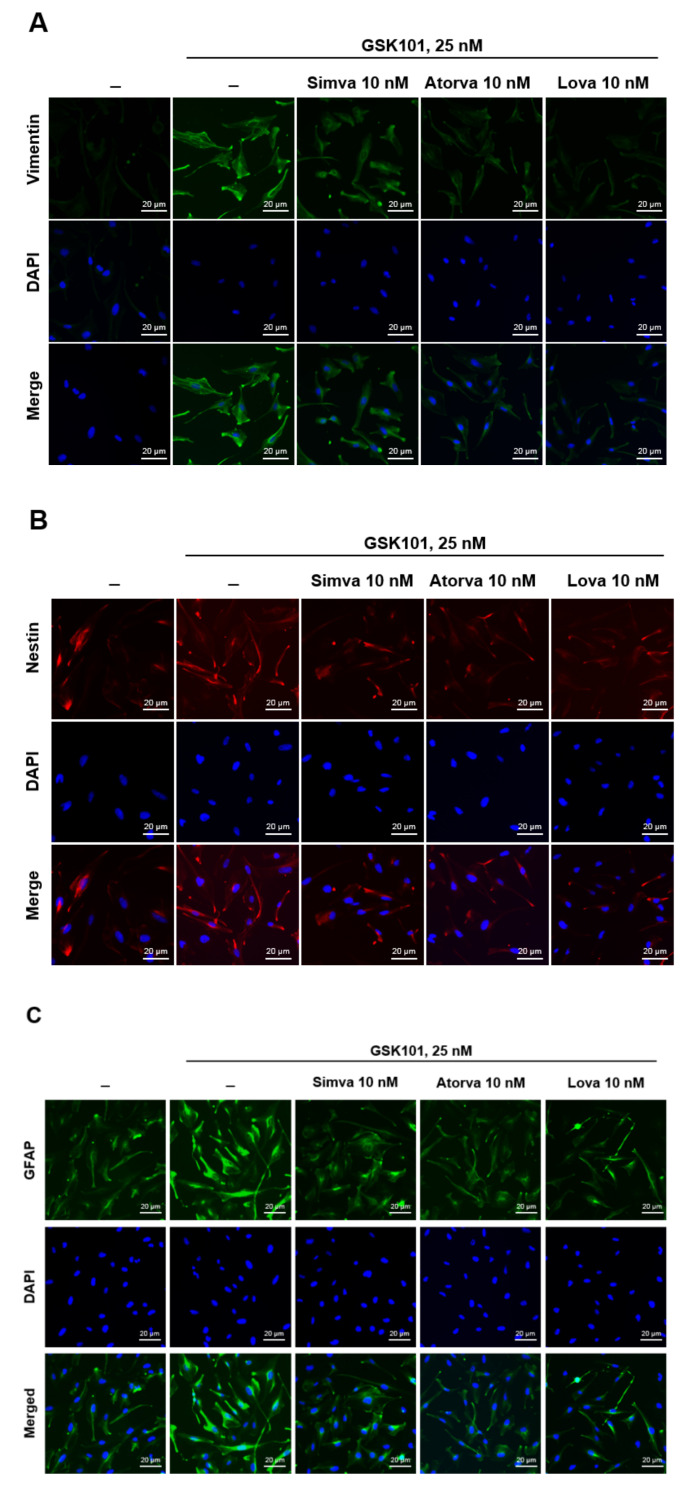
GSK treatment induces the reactivation of MIO-M1 cells and gliosis, and statins suppress these events. MIO-M1 cells were treated with GSK101 alone for 24 h or pretreated with each statin for 6 h, followed by GSK101 treatment for 24 h. Then, cells were stained for the expression of (**A**) vimentin, (**B**) nestin, and (**C**) GFAP, which are the indicators of gliosis. The nucleus was labeled with DAPI. GSK101 at 25 nM increased vimentin, nestin, and GFAP expression. Statins pretreatment at 10 nM attenuated these expressions. *n* = 4 for each figure.

**Figure 2 ijms-23-05190-f002:**
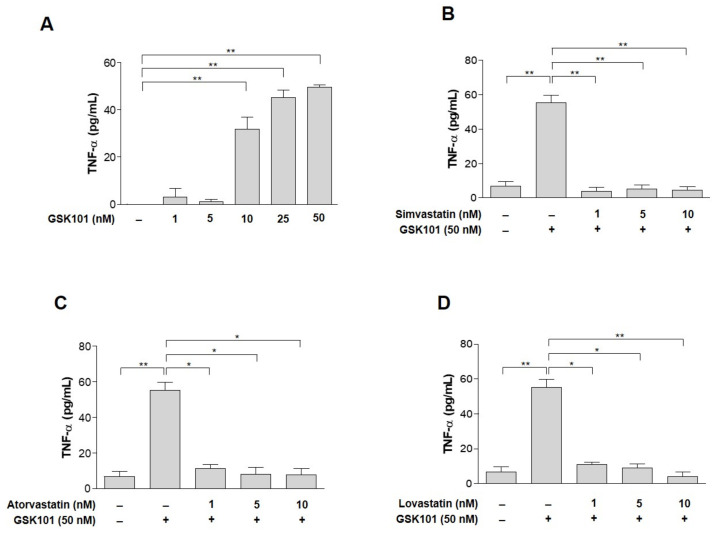
GSK treatment induces TNF-α and pretreatment with statins suppress it. TNF-α production was 14-fold greater than at a 1 nM concentration (**A**). GSK101-mediated induction of TNF-α levels suppressed by pretreatment with statins. MIO-M1 cells were treated either with GSK101 alone for 24 h or pretreated with (**B**) simvastatin, (**C**) atorvastatin, or (**D**) lovastatin for 1 h and then followed by GSK101 treatment for 24 h with 50 nM GSK101. Pretreatment with statins suppressed TNF-α production from 1 nM concentrations. Asterisks indicate statistical significance. * *p* < 0.05, ** *p* < 0.01. *n* = 4 for each figure.

**Figure 3 ijms-23-05190-f003:**
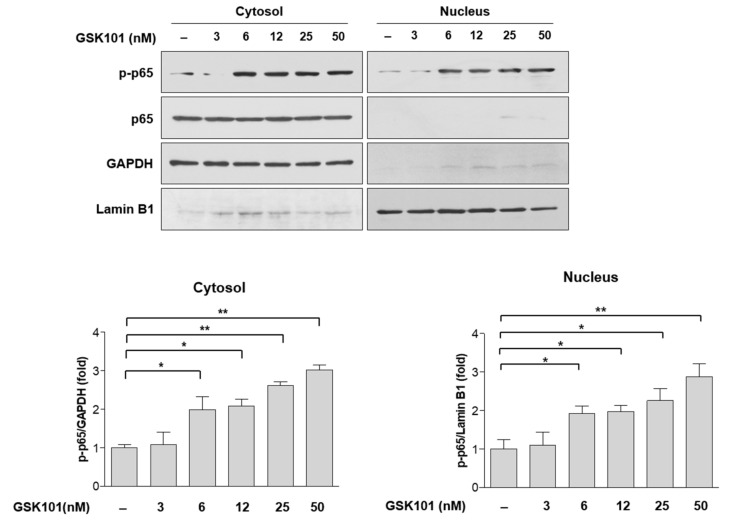
Phospho-p65 levels were induced by GSK101 in primary Müller cells. MIO-M1 cells were treated with increasing does of GSK101 for 24 h. Cell lysates were fractionated into nucleus and cytosol fractions and then subjected to Western blot. From 6 nM GSK101, p65 phosphorylation was significantly induced in both the cytosolic and nuclear fractions in a dose-dependent manner. GAPDH and Lamin B1 were served as the internal controls for cytosol and nucleus fractions, respectively. Asterisks indicate statistical significance. * *p* < 0.05, ** *p* < 0.01. *n* = 4.

**Figure 4 ijms-23-05190-f004:**
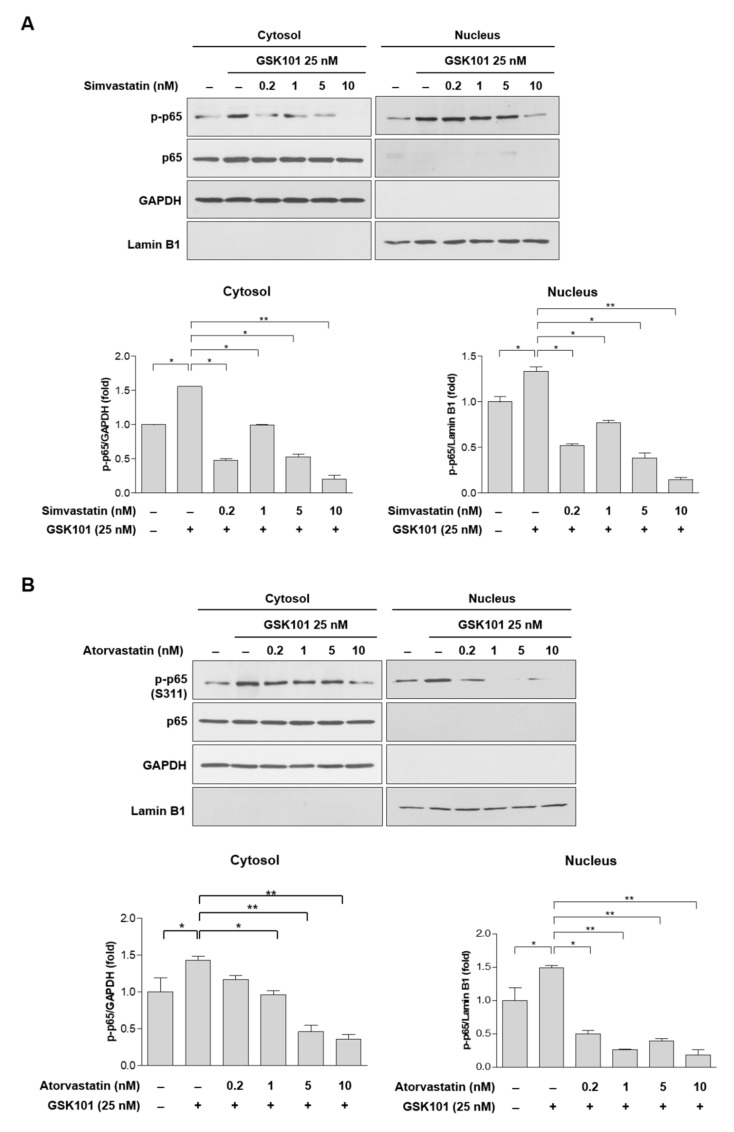
GKS101-mediated phosphorylation of p65 attenuated by statins. MIO-M1 cells were treated either with GSK101 alone for 24 h or pretreated with (**A**) simvastatin, (**B**) atorvastatin, or (**C**) lovastatin for 6 h, followed by GSK101 treatment for 24 h. The cytosol and nucleus fractions were subjected to Western blot analysis. P65 phosphorylation significantly decreased in the nucleus after pretreatment with 0.2 nM of all three statins and decreased in the cytosol after pretreatment with 0.2 nM simvastatin, 1 nM atorvastatin, and 10 nM lovastatin. GAPDH and Lamin B1 were served as the internal controls for cytosol and nucleus fractions, respectively. Asterisks indicate statistical significance. * *p* < 0.05, ** *p* < 0.01. *n* = 4 for each figure.

**Figure 5 ijms-23-05190-f005:**
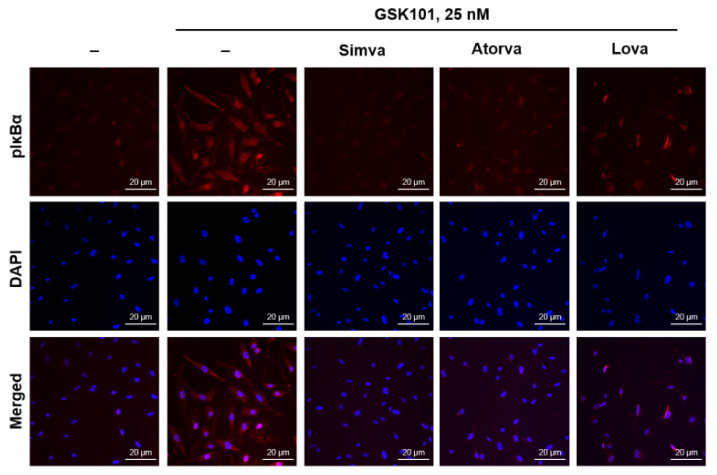
Statins attenuate GSK-mediated IκBα phosphorylation, which occurs upstream of p65 activation. MIO-M1 cells were treated either with GSK101 alone for 24 h or pretreated with simvastatin, atorvastatin, or lovastatin for 6 h, followed by GSK101 treatment for 24 h. Then, cells were fixed and stained for pIκBα. Pretreatment with 10 nM statins prior to GSK101 treatment suppressed IκBα phosphorylation. All three statins were potent attenuators of GSK-mediated IκBα phosphorylation upstream of p65 activation in Müller glial MIO-M1 cells. *n* = 4.

## Data Availability

Not applicable.

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
