# Peer review of "Statins Inhibit the Gliosis of MIO-M1, a Müller Glial Cell Line Induced by TRPV4 Activation"

_ijms, 2022, doi:10.3390/ijms23095190_

Round 1
Reviewer 1 Report
In this manuscript, the authors describe muller cell gliosis induced by TRPV4 activation and statins can mitigate gliosis using human muller cell line. In addition, pretreatment with statins has been found to inhibit gliosis, TNF-a production, IkBa phosphorylation, suggesting involvement of NF-kB pathway. The authors’ work may provide basic information on how muller cell gliosis occurs and is regulated by signaling pathway, and how statins work. Overall, the manuscript is interesting, informative and well-written. The experiment is well designed, and the interpretation of the data is appropriate. However, I have some minor suggestions to strengthen the manuscript.
Minor:
- It may be necessary to describe the number of samples and repeating for each figure.
- The discussion is quite simple that it would be nice to build up more.
Author Response
- It may be necessary to describe the number of samples and repeating for each figure.
- Thank you for the good comment. We added the number of samples and repeating for each figure.
2. The discussion is quite simple that it would be nice to build up more.
- Thanks for the good comment. We added the mechanism statin and limitations in discussion section. (2nd paragraph of discussion section)
Reviewer 2 Report
The manuscript ‘Statins inhibit gliosis of MIO-M1, a Müller glial cell line, induced by TRPV4 activation’ attempts to provide new insight into the impact of statin treatment on gliosis in glial cells. The text is rather well organized and the results are of interest.
However, I have several minor concerns:
- Add more details to captions for figures (e.g., time of incubation)
- How do authors explain the direct link between statins and the analyzed parameters (e.g. vimentin, nestin, GFAP, TNF-α expression). Is it due to the loss of the isoprenoids used for prenylation or cholesterol depletion (as discussed in DOI:1021/acs.jmedchem.1c00410)
- Discuss the literature where statins are ineffective in reducing gliosis (e.g. DOI: 1099/jgv.0.000876)
- Remove some typos., e.g.
Treatment with the TRPV4 agonist GSK101 upregulated vimentin, nestin, and GFAP expression (Fig. 1A, B, C) Pretreatment
- Use authors names instead of ref, e.g.,
Also, [46] reported that simvastatin may stabilize visual field progression.
Author Response
- Add more details to captions for figures (e.g., time of incubation)
- Thank you for the good comment. We added more details to captions for figures.
2. How do authors explain the direct link between statins and the analyzed parameters (e.g. vimentin, nestin, GFAP, TNF-α expression). Is it due to the loss of the isoprenoids used for prenylation or cholesterol depletion (as discussed in DOI:1021/acs.jmedchem.1c00410)
- Thanks for the great comment. It is thought to be due to decrease in isoprenoids used for prenylation. Reduction of isoprenoid inhibits the RhoA pathway, which in turn appears to decrease gliosis. Aizahn et al. reported that Rho kinase inhibition suppressed gliosis.(https://doi.org/10.1159/000350138) This content has been added to the discussion. (2nd paragraph of discussion section)
3. Discuss the literature where statins are ineffective in reducing gliosis (e.g.DOI: 1099/jgv.0.000876)
- Thank you for the good comment. We added that literature to the limitation section within the discussion.
4. Remove some typos., e.g.
Treatment with the TRPV4 agonist GSK101 upregulated vimentin, nestin, and GFAP expression (Fig. 1A, B, C) Pretreatment
- Thank you for the good comment. We fixed typos.
5. Use authors names instead of ref, e.g.,
6. Also, [46] reported that simvastatin may stabilize visual field progression.
- Thank you for the good comment. We add the names of the authors.